# Detection of Electrical Circuit in a Thin-Film-Transistor Liquid-Crystal Display Using a Hybrid Optoelectronic Apparatus: An Array Tester and Automatic Optical Inspection

**DOI:** 10.3390/mi12080964

**Published:** 2021-08-15

**Authors:** Fu-Ming Tzu, Jung-Shun Chen, Shih-Hsien Hsu

**Affiliations:** 1Department of Marine Engineering, National Kaohsiung University of Science and Technology, Kaohsiung 80543, Taiwan; 2Department of Industrial Technology Education, National Kaohsiung Normal University, Kaohsiung 80201, Taiwan; jschen@nknu.edu.tw; 3Department of Electrical Engineering, Feng Chia University, Taichung 40724, Taiwan; shihhhsu@fcuoa.fcu.edu.tw

**Keywords:** array tester, optical inspection, time delay integration charge-coupled device (TDI-CCD), thin-film-transistor liquid-crystal display (TFT-LCD), optoelectronic apparatus

## Abstract

In this study, we developed a high-resolution, more accurate, non-destructive apparatus for refining the detection of electrode pixels in a thin-film-transistor liquid-crystal display (TFT-LCD). The hybrid optoelectronic apparatus simultaneously uses an array tester linked with the automatic optical inspection of panel defects. Unfortunately, due to a tiny air gap in the electro-optical inspector, the situation repeatedly causes numerous scratches and damages to the modulator; therefore, developing alternative equipment is necessary. Typically, in TFT-LCDs, there are open, short, and cross short electrical defects. The experiment utilized a multiple-line scan with the time delay integration (TDI) of a charge-coupled device (CCD) to capture a sharp image, even under low light, various speeds, or extreme conditions. In addition, we explored the experimental efficacy of detecting the electrode pixel of the samples and evaluated the effectiveness of a 7-inch opaque quartz mask. The results show that an array tester and AOI can detect a TFT-LCD electrode pixel sufficiently; therefore, we recommend adopting the hybrid apparatus in the TFT-LCD industry.

## 1. Introduction

The most popular display technology is that of the thin-film-transistor liquid-crystal display (TFT-LCD), accounting for most global display markets [1]. Lightweight, high-resolution, low-power consumption, low-temperature fabrication processes, and multiple other functions have become increasingly prominent among different types of display [2,3]. However, as the hundreds of millions of transistors shrink inside the minuscule electrode pixel, the electrical pattern becomes complex. Furthermore, since the electrical current function dominates the transferring, any abnormal signal delivery may severely damage the circuit loop; therefore, a high-resolution, more accurate, and non-destructive detection apparatus is necessary. An array tester focuses on the electrical signal of the active area on a TFT-LCD to detect the defect of the metal line and to identify the effectiveness of the subpixel, as well as to simultaneously classify the defect types for other repair processes. 

The primary electrical tester of TFT-LCDs utilizes an electro-optical inspector [4] and an array tester [5]. The electro-optical inspector is the most popular technique for detecting the in-cell flawed circuit pattern of a TFT-LCD. A modulator applies an electric field to induce the electrical flow and turn on the liquid crystal on the surface of the electrode pixel. However, a gap of approximately 20~50 μm exists between the modulator and the sample, and the inspector covers the whole area during testing. In the case of foreign material adhering to the example, the condition frequently damages the modulator of the electro-optical inspector. The statistical results of one study have indicated 70% fragmented glass, 20% metal particles, and 10% other foreign materials in a TFT-LCD factory [6]. Broken glass during production is inevitable due to mishandling by the operator, sensor problems of the robotics, etc.

The array tester is a signal point to detect the gate and data line in the TFT-LCD. Its probe provides an excellent fine pitch to contact the external array of TFT-LCD and detect the abnormal signal while the integrated circuit (IC) is turned on. Thus, the damage risk is low and maintains ease for the probe. In addition, the measured speed is significantly faster. Moreover, the array tester also applies to the wafer and electronic package of the semiconductor test. However, small dimensions, high definition, and pixel density in the electrode present more challenges. 

Previous literature reviews have described studies on the related defect issues. For example, Oh et al. (2010) [7] utilized optical components of a scanning electron microscope (SEM) with a 7-inch TFT-LCD panel to inspect a thin-film transistor with some intentionally made defects such as open data or gate lines by cutting some points using a laser beam. However, the e-beam confined the operation due to its high cost, low throughput, and possible damage due to high e-beam energy. Although the result was significant, the process was complex and operated under high vacuum conditions, i.e., at 10^−^^7^–10^−^^8^ Torr. 

Tsutsumi et al. (2018) [8] a fast optical inspection combined with a multi-zoom visual ability in a microscope that simultaneously enabled visualization and review. Using gate modulation (GM) imaging measurements, the technique operated up to 30,000 pixels in the field of view for large-area active-matrix (AM) backplanes. The method was very impressive for observing the defect by the GM photosensor. The study focused on optical magnification and multi-zoom, but the electrical edge probe did not utilize detection in the experiment. 

Zhang et al. (2019) [9] explored the distance measuring the optical autofocus at the area photosensor based on laser triangulation. The result satisfied an industrial TFT-LCD in the active region. Moreover, the auto-focus observation results were impressive in terms of inspection. Nevertheless, the detective system needed multiple aspects to consider key issues such as the design architecture and gray level of the binary technique. 

Lee et al. (2020) [10] performed an active contour model (ACM) with image resampling on a thin-film transistor to measure the TFT-LCD pad pattern critical dimension at a width of 13 microns. They focused on an edge and gray level algorithm of the subpixel to detect the subpixel. The authors utilized edge extraction by pixel-level edge detection, pixel classification, image force fine pixel resampling, and subpixel level edge detection with ACM. The image was constructed by a filter kernel using a 3×3 matrix. The result was very significant, especially in terms of the kernel software, for improving the gray level and edge algorithm. However, the software program is one factor to consider in defect inspection compared with the optical parameter and optoelectronic apparatus. This study developed a high-resolution line-scan image to implement a defect-free process to satisfy an extensive display. 

Therefore, we developed an optoelectronic apparatus with an array tester linked with automatic optical inspection (AOI) to detect the electrical signal on the electrode pixel of the TFT-LCD array. In this study, we experimentally investigated the circuit loop of open, short, and cross short defects on the electrode pixel and evaluated the optical resolution using a 7-inch photomask. As a result, the developed apparatus exhibited a damage-free array tester and accurate quantitative statistics. 

## 2. Principles of Array Tester Linked with Automatic Optical Inspection

The morphology of the defects in the TFT-LCD has a diversified type. Notably, the color distortion appears in the display in the case of an abnormal electrical switch. For example, a malfunction of the signal transmission on the electrode pixel causes abnormal liquid crystal rotation between the TFT and the color filter. Therefore, the experiment specifies a lethal threat for evaluating open, short, and cross short defects using an array tester linked with AOI to quantitatively detect flaws on the electrode pixel. 

Furthermore, the essential AOI utilizes gray level distribution by line scan to recognize a set of flaws for open, short, and cross short defects. The gray level of an image is a compelling method to acknowledge a photo defect. The grayscale was obtained by measuring the brightness of each pixel in a single electromagnetic spectrum analogy with monochromaticity. The digital process has 256 grayscales in the evaluation. The captured picture on the electrode of the TFT-LCD array transforms the image from the gray-level to binary threshold to simplify the intricate image. Every line defect is effortlessly recognized through multiple exposures to compose the sharpest image experimentally. The optical inspection is based on a span of gray level from 0 to 255. In the evaluation, the gray image of the threshold *m* is set to the criterion of binarization for the digital image process; therefore, Equation (1) expresses the image grayscale as follows [11,12]: (1)$$m=\sum_{i=1}^{n}f\left(x,y\right)$$
where *f* is the process that captures the photo and *n* represents the pixel quantity in the image.

This is a fundamental element to construct the photo. In the experiment, the pixel’s dimension depends on the optical resolution and the brightness in terms of light intensity. The brightness of the pixel determines the grayscale in the image. The summation is from 1 to *n*. The gray value is set to *f(x, y)* concerning *(x, y)*. In addition, if the gray value is lower than the *m* of the grayscale, the binary image is set to 0. If the image of the gray value is higher than the criterion of the *m*, the binary image is set to 1. Thus, the binary image display is black and white, two-tone, and monochromatic. 

The experiment handles the image based on the captured picture by scanning. First, the image is acquired by time delay and the integration of the charge-coupled device (TDI-CCD). Second, the experiment sets a reference basis and then computes the highest gray pixels of the image. As a result, the unique average gray pixel may be highlighted among the photos. 

The gray level distribution is set for the method by subtraction, as shown in Figure 1. Thus, the defect in the image appears obvious. The captured image transfers the computer process system to the automatic data acquisition program (ADAP).

Time-delay integration (TDI) is the most efficient among line-scan photosensors [13,14]. The scanning capacity is the fastest progressive photosensor with a rectangular shape manufactured by Teledyne Dalsa, Waterloo, Ontario, Canada. The heightened sensibility (HS) of a 12K TDI-CCD has a line rate of 96 kHz and an optical resolution of 5.2 μm. The traditional line scan detects the objective with one scan at a time compared with the TDI-CCD configuration, shown in the array to capture images with multiple scans at one time, which is more potent than a traditional line scan. Moreover, the TDI-CCD can charge the electronic signal along the motion axis to accumulate a multi-stage image. The packets of the signals gradually increase the datum to amplify the information through time-delay integration. While the sensor delivers the picture, the signal storage is proportional to the exposure time. Thus, the TDI-CCD illustrates a higher sensibility than transitional linear CCD and captures the image under low light conditions. Cumulative exposure performs forward and reverses scanning, and the transmission of the camera-link with a heightened sensibility (HS) interface is at a 256 gray level. At the same time, the photosensor outperforms this under a weak light source, even at a slow velocity, and grabs a picture of a moving object during the transfer of the synchronous charge scanning. Therefore, it demonstrates the characteristics of high speed and high sensitivity without a vital light source. 

Optical detection utilizes a non-contact method to detect the metal line of TFT-LCD. Thus, the optical solution becomes an essential issue to capture the image. In machine vision, the working distance expresses the height between the photosensor and the sample that decides the sharpness of the picture. The experiment takes the following equation to design the optical measurement as follows [15,16]: (2)$$WD=\left(\frac{M+1}{M}\right)+f\times \left(M+1\right)+\bar{HH^{*}}$$*f* is the focal point, and *M* expresses the magnification between the object and subject and the $\bar{HH^{*}}$ lens’s thickness. 

Moreover, the depth of focus (*DoF*) depends on the optimum sharpest image, a critical parameter. Thus, *DoF* is relatively short-range before and after the height [17]. The object distance depends on the visual height, aperture diameter, and focal length. If the image is inside, the height of the *DoF* is clear. In contrast, if the object located is too close or too far, then the *DoF* is blurry. Equation (3) presents the formula as follows: (3)$$DoF=\frac{sf^{2}}{f^{2}-Ac(s-f)}-\frac{sf^{2}}{f^{2}+Ac(s-f)}$$

The parameter expresses the subject distance, *f* indicates a focal length, *A* is an aperture number, and factor *c* is the circle of confusion. 

In the optical field of view, the size (area) of the CCD is a significant component to perform the characteristic links with the magnification lens. Usually, the lens can magnify from 1×, 2×, 5×, 10×, 20×, and 50× and above; however, high magnification means a large view and vice versa. Figure 2 illustrates the optical path with the field of view at the area CCD. Figure 2A demonstrates the various sizes of the CCD and draws the width × height (W × H, mm × mm). Figure 2B illustrates the optical path to magnify the object. Equation (4) of the field of view (FoV) is as follows [18,19]: (4)FoV=resolution×pixel

The field of view presents the optical resolution times the total pixel of the photosensor. The pixel is the smallest unit to constitute an image dot matrix; the size depends on the optical resolution: a smaller pixel means higher resolution and vice versa. 

The detectability varies with an optical resolution in AOI. The inspection applies to micromachines, semiconductors, solar panels and various other applications. Typically, the optical inspection method consists of the photosensor, grabber card, analog–digital converter, magnification lens, and mechanism parts, including the moving gantry, server motor, and precise optical ruler that supports the accurate position. The architecture of the optoelectronic apparatus is presented in the following section. 

## 3. Architecture for Array Tester Linked with Automatic Optical Inspection

We developed an optoelectronic apparatus, a hybrid array tester linked with AOI, to detect open, short, and cross short defects on TFT-LCD electrode pixels. Figure 3 shows the topology of the array tester linked with AOI, which utilizes the external electrical probe contacts at the edge along with the array. Consequently, the edge inspection of the array tester detects the electrical signal. While the readily inspected glass enters Station 1, the platform provides a vacuum such that the glass is as smooth as possible. The glass is in the proper position and is ready to be delivered to Station 2 for the array tester, optical inspection, and review of the captured image. Then, Station 3 provides the glass-out.

Furthermore, the experiment detects the electrical defect on the electrode pixel of the TFT-LCD of the 8.5th generation. The dimensions of the glass are 2500 mm × 2200 mm, and it was manufactured by Corning Inc. (Corning, NY, USA), with 10 class levels of clearing room. Fukan Co., Ltd. (Fukuyama, Hiroshima, Japan) produced the array tester, which is fast, accurate, and performs quantitative analyses at the shortest time of 3 ms. Due to the low-frequency vibration at the manufacturing site, the vibration-free granite material with 300 mm thickness is necessary. In addition, the air suspension of the platform is designed for pneumatic anti-vibration. While the magnified lens is installed on the photosensor, the TDI-CCD that captures the image is more precious, utilizing a commercial-off-the-shelf product—that of a multiple progressive line scan. As a result, the TDI-CCD provides quick responsivity under weak optics and slow velocity. Thus, we intend to develop an alternative solution, a hybrid electrical inspection with an external array tester engaged with multiple line scans. 

## 4. Results and Discussion

The investigation involved several experiments to evaluate the possibility of an optoelectronic apparatus comprised of an array tester linked with the automatic optical inspection. First, we investigated the foreign material damage of the modulator during production. The topology of the electro-optical inspector is illustrated in Figure 4; Figure 4A shows the foreign material adhered to the modulator, and Figure 4B illustrates the practical modulator in the TFT-LCD factory. The experiment analyzes a damaged modulator by optical detection. The evaluation was at a bevel angle of 30 degrees between the line-scan CCD and light source at an optical resolution of 5.2 μm, as shown in Figure 5. The assessment of the damaged modulator is shown in Figure 5A–C; Figure 5A illustrates the topology of the optical defect, Figure 5B demonstrates a broken modulator at 143 mm × 133 mm, and Figure 5C shows the scratch defects detected in the image. 

In the electrode pixel of the TFT-LCD, the electrical circuit consists of the metal’s lines, i.e., the data line and the gate line. The current drives the electronic signal to rotate the liquid crystal and adjusts the chromaticity. Thus, each pixel has two metal lines across the region. The data line determines the voltage that each pixel needs to reach its brightness. The gate line controls the electrical current in the alignment layer to decide the screen resolution. However, once the current encounters an obstruction that blocks the signal in the loop, the electrode pixel shows a malfunction. An open defect is when no current passes in the metal line. This means that the sensor cannot receive an output voltage, as shown in Figure 6; Figure 6A illustrates the array tester, Figure 6B indicates the probing on the glass at the open circuit, and Figure 6C shows that the output voltage has no signal, indicated by the red circle in the sensor numbers 3~4. Secondly, the short defect condition indicates that each sensor receives twice the expected output voltage. Therefore, the output voltage is two times higher, as shown in Figure 7; Figure 7A illustrates the array tester, Figure 7B indicates the probing on the glass at the short circuit, and Figure 7C illustrates two times the output voltage, as indicated by the red circle. Thirdly, in the cross short defect states, the sensor receives a higher voltage than the predicted output, and therefore the output voltage is more elevated, as shown in Figure 8; Figure 8A illustrates the array tester at the cross short, Figure 8B indicates the probing on the glass at the cross short circuit, and Figure 8C indicates a voltage tendency of the sensor, shown as a red circle in the sensor numbers at 3~4 [20]. 

The experiment inspects the 46 pieces of the 8.5th generation glass for open, short, and cross short defects. As a result, the statistical measurements illustrate the tendency of the circuit characteristics by the array tester, as shown in Figure 9 (the short defect in Figure 9A; cross short defect in Figure 9B; open defect in Figure 9C; and the regular circuit in Figure 9D). The tendency is the same as previously shown in Figure 7, Figure 8 and Figure 9. The results indicate that the short defect (Figure 9A) voltage is two times higher than that of the regular circuit (Figure 9D). The cross short defect (Figure 9B) shows a peak voltage at sensors no. 17~19 over regular circuits. The open defect (Figure 9C) exhibits the jump to interrupt the voltage at sensors no. 17~19, and no current passes in the end. 

The experiment also verified the non-contact optical method of detecting an electrical pattern. Sharp vision detects the flaw morphology; the defects include electrical-related defects and also non-uniformity morphology. The evaluation utilizes a standard pattern to perform the optical capability, an opaque quartz mask concerned with a 7-inch diagonal, and dimensions of 177.8 × 177.8 × 3.2 (L × W × T, unit mm). The standard pattern is manufactured by JD Photo Data Inc. (Hitchin, Herts, UK). Figure 10 illustrates the whole detailed area of the quartz mask that inspects the TDI-CCD line-scan compatibility. The design of the pattern of the quartz mask consists of (1) a normal area; (2) a defect area; (3) a metal layer; (4) a test key; and (5) other. We focused on the (2) defect area for evaluation, which means that the central area consists of the normal, short, open, pinhole, and island types. The defect size in the descending order is 20, 10, 9, 8, 7, 6, 5, 4, 3, 2.5, 2, 1.5, 1, and 0.5 μm, as shown in Figure 11. The defect consists of open and short electrical patterns in the outsize region and the peripheral pattern. 

We utilize the high resolution of the TDI-CCD photosensor to scan the 7-inch quartz mask that verifies detectability, as shown in Figure 12. The blue circle consists of four defects and is ready for detection: two short defects and two open defects. Consequently, the red arrow illustrates a clear photo. In Figure 13, the blue circles exhibit the defects in the picture; a similar noodle shape is dispersed, but the TDI-CCD also detects short and open defects. Finally, Figure 14 indicates the other area in the 7-inch quartz photomask that is an island defect. 

The experiment captured the image that depends on several optical parameters such as the working distance (WD), depth of focus (DoF), and field of view (FoV). Table 1 tabulates the CCD pixel size at the optical resolution of 5.2 μm, with the 10× lens to magnify the object with 0.52 μm resolution. The focal length refers to the distance from the center of the lens to the photosensitive element. Each lens has a different focal length. Therefore, the focal length plays a vital role in determining the field of view of the shot.

Consequently, a shorter focal length means a more comprehensive shooting range, whereas a longer focal length means that distant objects appear larger. Ordinarily, a focal length of 35 mm is a standard lens because its angle of view is close to that of a human. Thus, a wide-angle lens has a focal length lens shorter than 35 mm, and a telephoto lens has a longer focal length. However, the experiment chose the 20 mm focal length with a wider-angle field and a higher optical resolution. 

The depth of focus (DoF) means the maximum range at which an object can be appropriately focused upon, illustrated by Equation (3). In addition, the aperture adjusts the entering light when taking pictures. Therefore, *F/X* indicates the size of the aperture. Since the *F* value fixes, the smaller the “aperture value” as the denominator is, the greater the amount of entering light is, and the shallower the depth of the field; the aperture value is more significant; and the smaller the light input is, the deeper the focus is—as shown in Table 2 [21]. 

The field of view (FoV) indicates the whole area to observe and images the photosensor by the lens, shown in Table 3. The evaluation takes the various magnified lenses of 1×, 2×, 5×, 10×, 20×, and 50×, the color temperature of 6500 K, the 30-frame rate of the area CCD, and captures the island defect electrode pixel in the TFT-LCD—as shown in Figure 15. Thus, the electrode circuit’s zoom-in photo shows higher magnification, better resolution, and a refined image. At the same time, the magnification ratio indicates a relationship between the sensor size and FoV. In contrast, the one-inch diagonal of the sensor dimension is carried out in the experiment. 

We developed a hybrid architecture of the optoelectronic apparatus using an array tester linked with the automatic optical inspection. The statistics of the electrical profile and line-scan image are significant. Furthermore, the optical parameters of the working distance (WD), the depth of field (DoF), and the field of view (FoV) were performed in the evaluation to magnify the image resolution. 

## 5. Conclusions

In this study, our investigation shows the significance of open, short, and cross short defects on an electrode pixel of the thin-film-transistor liquid-crystal display using an array tester linked with automatic optical inspection. First, the experiment takes a photosensor to detect the damaged modulator. The result was undeniable due to the tiny air gap of the electro-optical inspector. Later, the array tester can identify the various voltage profiles of online samples using an excellent miniature pitch probe. Consequently, the open, short, and cross short defects indicate an evident tendency. The open defect exhibits the interruption of the voltage, and no current passes in the end. The voltage of the short defect is two times higher than that of the regular circuit. The cross short defect shows a peak voltage more significant than typical circuits. Furthermore, an optical inspection captures the image with better accuracy and high resolution using the TDI-CCD photosensor, verifying the detectability with the optimized optical parameter using an opaque quartz mask, considering the working distance, depth of focus, and field of view. Thus, the evaluation utilizes the hybrid optoelectronic apparatus to successfully detect electrical defects on the electrode pixel.

## Figures and Tables

**Figure 1 micromachines-12-00964-f001:**
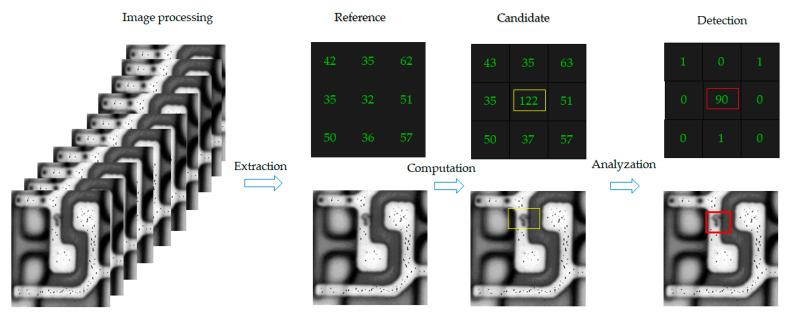
The acquired image is extracted, computed, and analyzed by automatic data acquisition program (ADAP).

**Figure 2 micromachines-12-00964-f002:**
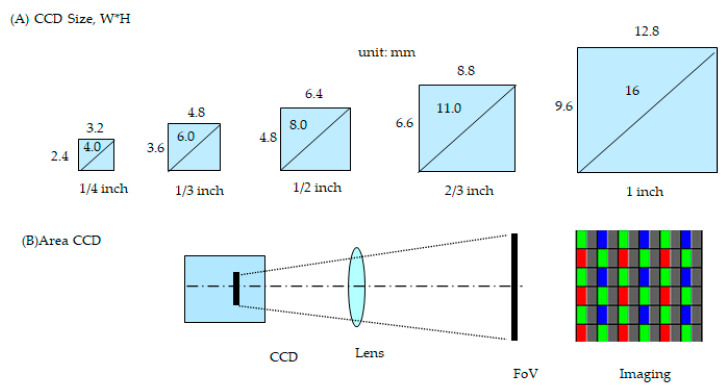
Illustration of various sizes of charge-coupled devices (CCDs) from small to large at the field of view: (**A**) various CCDs; and (**B**) demonstrates an optical path with the field of view.

**Figure 3 micromachines-12-00964-f003:**
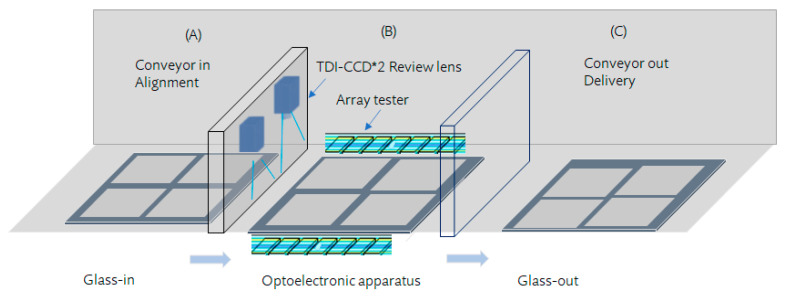
The architecture illustrates an array tester linked with the automatic optical inspection. (**A**) glass-in station; (**B**) optoelectronic apparatus station; (**C**) glass-out station.

**Figure 4 micromachines-12-00964-f004:**
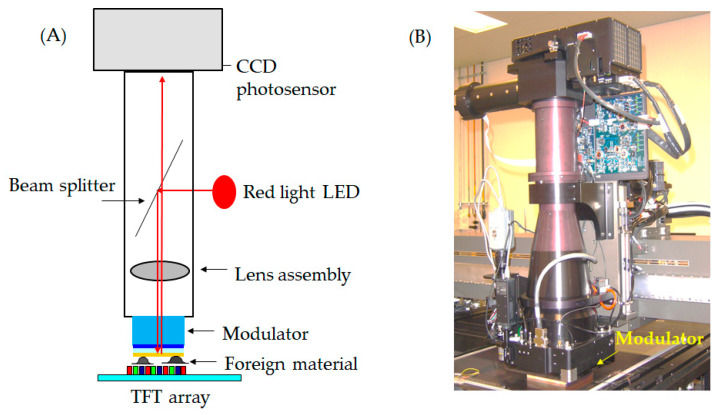
Topology of the electro-optical inspector demonstrates the function: (**A**) with adhered foreign material; and (**B**) the practical modulator in the TFT-LCD factory.

**Figure 5 micromachines-12-00964-f005:**
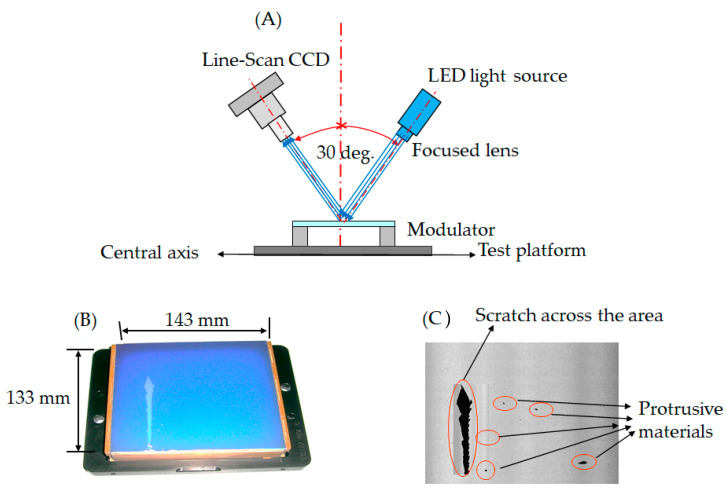
Topology of protrusion defects on the modulator: (**A**) architecture of machine vision; (**B**) a damaged modulator with the size 143 mm × 133 mm; and (**C**) gray image with visual defects.

**Figure 6 micromachines-12-00964-f006:**
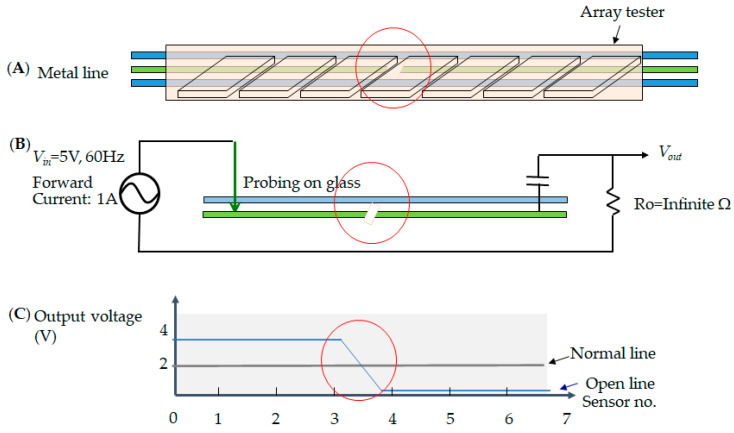
The open defect illustrates (**A**) the array tester on the glass; (**B**) that the sensor cannot receive an output voltage; and (**C**) a voltage tendency two times higher than the output voltage.

**Figure 7 micromachines-12-00964-f007:**
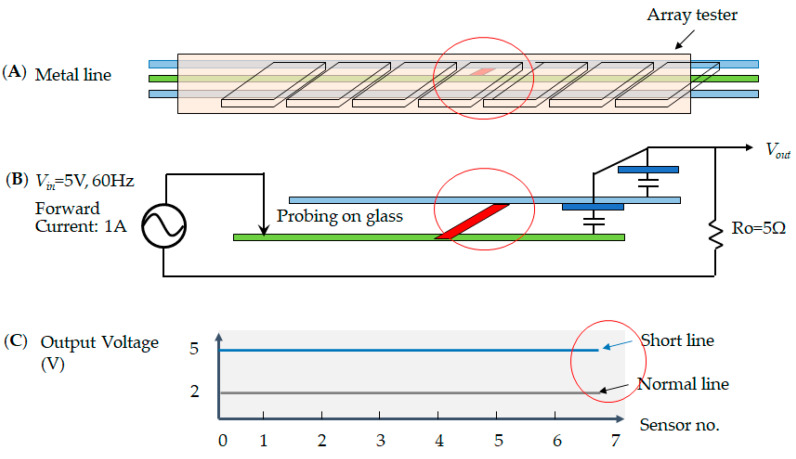
The short defect illustrates (**A**) the array tester on the glass; (**B**) the metal lines across one another; and (**C**) the at output voltage is two times higher than in the normal line.

**Figure 8 micromachines-12-00964-f008:**
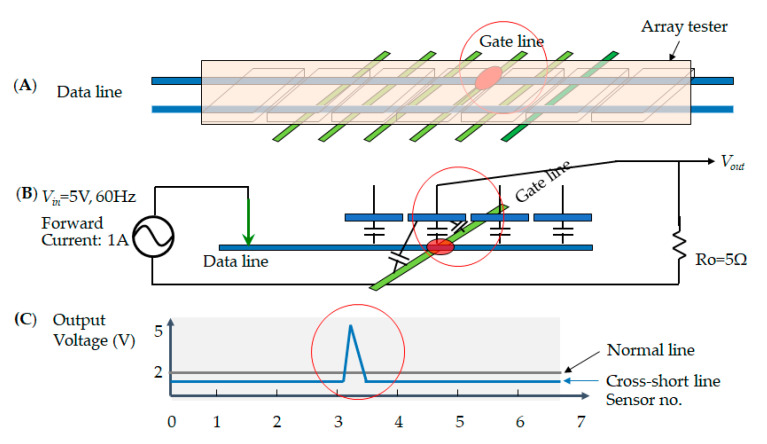
The cross short defect indicates (**A**) the array tester on the glass; (**B**) the electrical current with forwarding voltage; and (**C**) the voltage tendency of the sensor.

**Figure 9 micromachines-12-00964-f009:**
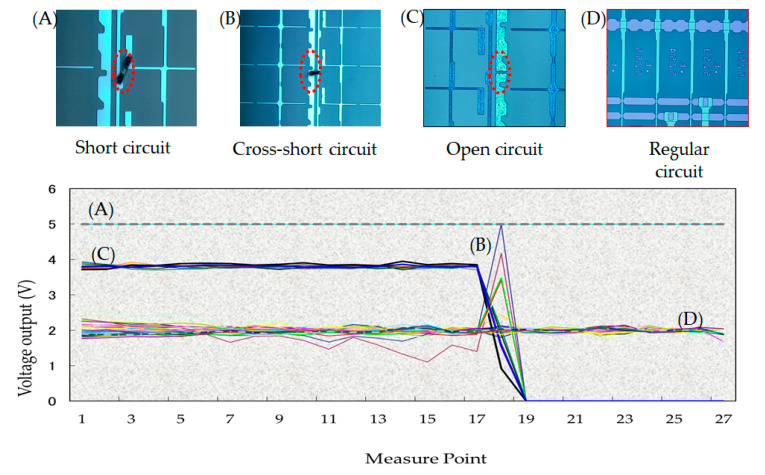
The voltage profile of the various conditions: (**A**) the short defect voltage; (**B**) the cross short defect voltage; (**C**) the open defect voltage; and (**D**) the regular circuit.

**Figure 10 micromachines-12-00964-f010:**
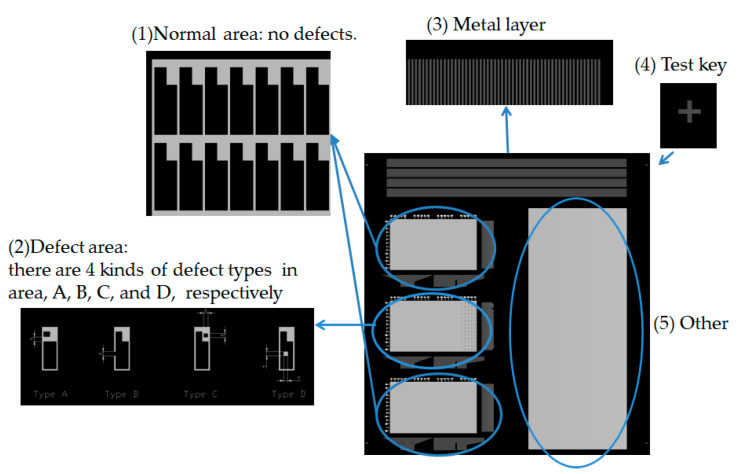
The architecture of the standard quartz mask is ready for line-scan detectability: (1) the normal area; (2) the defect area; (3) the metal layer; (4) the test key; and (5) other.

**Figure 11 micromachines-12-00964-f011:**
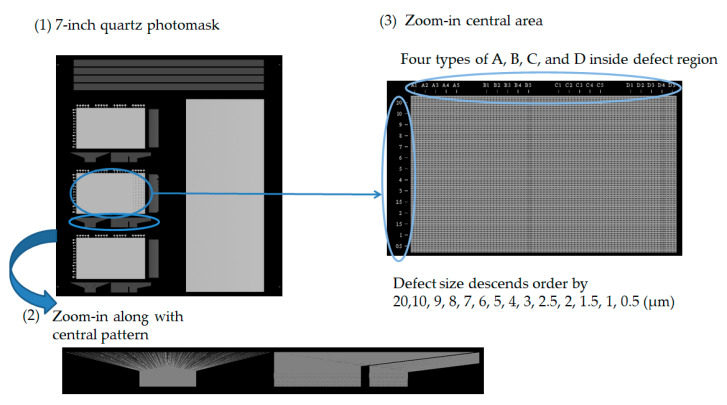
Illustration of the central area for a 7-inch quartz mask: (1) a 7-inch quartz pattern; (2) a zoom-in along with the central pattern; and (3) the zoom-in on the central area.

**Figure 12 micromachines-12-00964-f012:**
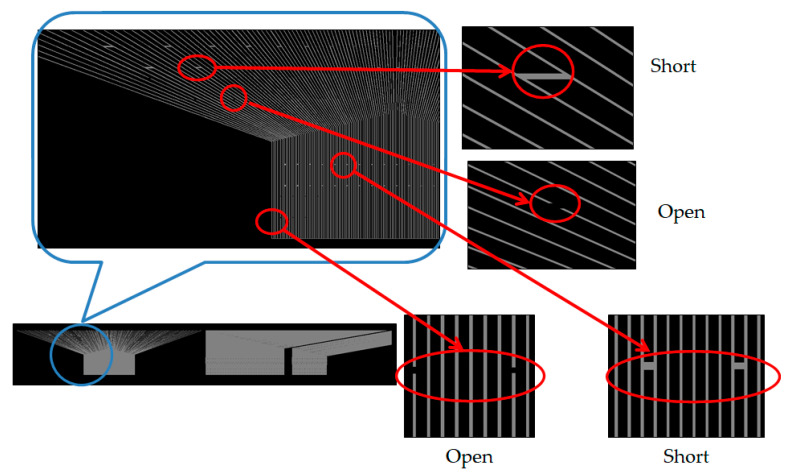
Illustration of the line scan at open and short defects on the left side of the photomask.

**Figure 13 micromachines-12-00964-f013:**
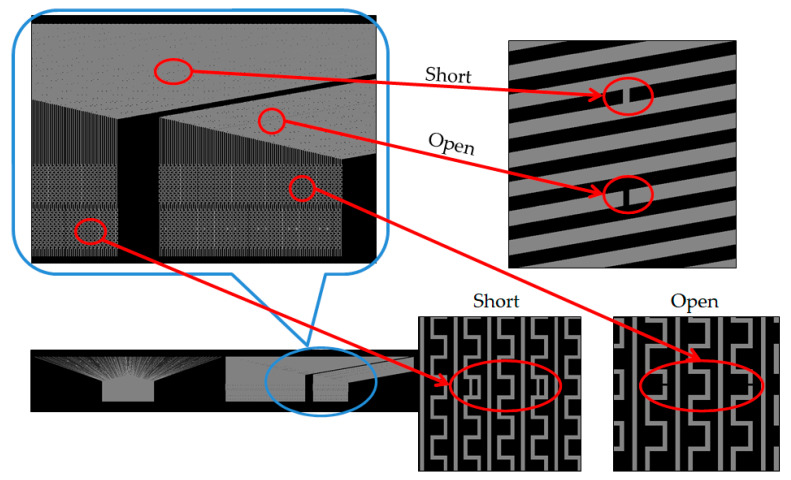
Illustration of line scan at open and short defects along with the peripheral right site of the photomask.

**Figure 14 micromachines-12-00964-f014:**
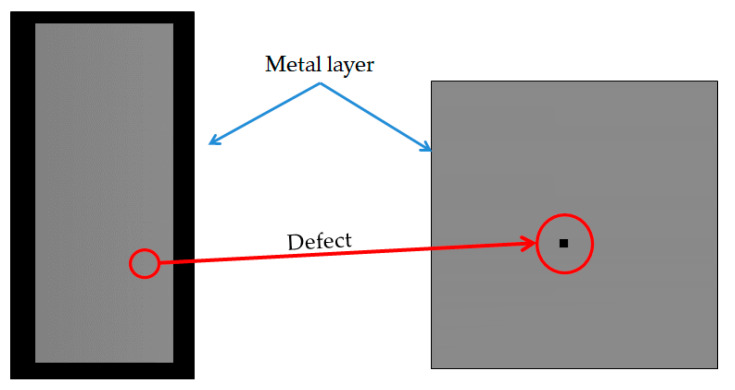
Illustration of line scan at the island/spot defect in another area of the photomask.

**Figure 15 micromachines-12-00964-f015:**
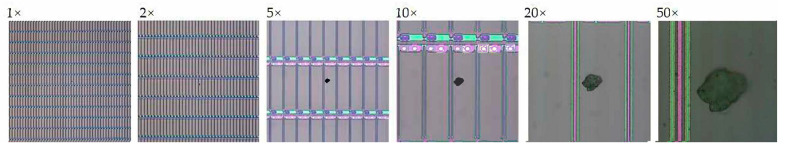
The captured photos by area CCD at various magnified lenses were 1×, 2×, 5×, 10×, 20×, and 50× at the island defect of the electrode in TFT-LCD.

**Table 1 micromachines-12-00964-t001:** The design of the working distance in the automatic optical inspection (AOI).

Parameter	Value	Unit
CCD pixel size	5.2	Micrometer (μm)
Magnification	10	None
Optical resolution	0.52	Micrometer (μm)
Lens focal length	20	Millimeter (mm)
Working distance	242	Millimeter (mm)

**Table 2 micromachines-12-00964-t002:** The design of the depth of focus in the AOI.

Parameter	Abbreviation	Value	Unit
Aperture number	*N*	2.4	None
Circle of confusion	*c*	0.012	None
Focal length	*f*	20	Millimeter (mm)
Subject distance	*s*	242	Millimeter (mm)
Depth of focus	*DoF*	7.5	Millimeter (mm)

**Table 3 micromachines-12-00964-t003:** The parameter of the field of view design in the area CCD.

Lens	CCD	Pixel Number		Resolution		FoV	
Magnification	Diagonal	W	H	W (μm)	H (μm)	W (μm)	H (μm)
1×	1″	1280	960	10	10	12,800	9600
2×	1″	1280	960	5	5	6400	4800
5×	1″	1280	960	2	2	2560	1920
10×	1″	1280	960	1	1	1280	960
20×	1″	1280	960	0.5	0.5	640	480
50×	1″	1280	960	0.2	0.2	256	192

## Data Availability

The data presented in this study are available on request from the corresponding author.

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
