# Peer review of "Detection of Electrical Circuit in a Thin-Film-Transistor Liquid-Crystal Display Using a Hybrid Optoelectronic Apparatus: An Array Tester and Automatic Optical Inspection"

_micromachines, 2021, doi:10.3390/mi12080964_

Round 1
Reviewer 1 Report
The paper is about an hybrid opto-electronic apparatus for the detection of fabrication defects (open, short and cross-short circuits) in a tunnel-film-transistor factory, in order to avoid damages during the detection.
I have read the entire paper, but, while the results of the paper seem interesting, unfortunately at the moment the text of the paper is impossible to judge because of the terrible English writing, which makes it incomprehensible (at least for me).
I suggest to the Authors to completely rewrite the text (with the aid of a person with good English skills) and to resubmit the new version for a new review round.
At this stage, I have suggested "major revisions" instead of "reject" only in the hope that the Authors will be able to present a strongly improved text.
Author Response
To Reviewer 1 response.
Point 1:
I suggest to the Authors to completely rewrite the text (with the aid of a person with good English skills) and to resubmit the new version for a new review round. At this stage, I have suggested "major revisions" instead of "reject" only in the hope that the Authors will be able to present a strongly improved text.
Answer 1: The authors express their appreciations for the reviewer's suggestion. The paper has undergone English language editing. The text has been checked for correct use of grammar and common technical terms by native English-speaking editors.

Reviewer 2 Report
The paper title “Detection of Electrical Circuit for Thin-film Transistor using Hybrid Optoelectronic Apparatus: Array Tester and Automatic Optical Inspection” authored by Tzu et al is an interesting experimental work on the hybrid opto-electronic apparatus, an array tester used in the simultaneous non-contact optical inspection and detects in the panel. However, the paper is suffering from poor English and errors in sentence formations. I highly suggest the authors improve the paper by taking Extensive English service to enhance the paper quality. Other than that I am willing to support its publication after minor corrections. Below are my few minor suggestions:
- AOI has been defined in line 91, don’t define it again in line 193. Use its abbreviated form directly.
- TFT has been defined in line 29, don’t define it again in lines 42 and 204. Kindly keep harmony with the abbreviation throughout the paper.
- The English used in the manuscript is very poor. It is difficult to understand most things. I suggest the authors do an Extensive sentence correction.
- Line 238, it's data line, not dateline. Correct.
- I am not able to understand why the author uses “which” in the explanations where he/she talks about figure A or B. For instance: “so the output voltage is twice higher in Figure 7, which figure 7 (A)”, “Figure 6, which figure 6 (A) illustrates the array tester”, “n Figure 4, which figure 4 (A) is the foreign material”. It is not correct. Need to perform an English correction.
Author Response
To Reviewer 2 response.
Point 1: I highly suggest the authors improve the paper by taking Extensive English service to enhance the paper quality.
Answer 2: The authors express their appreciation for the reviewer's suggestion. The paper has been consulted with support by MDPI publish, which has undergone English language editing. The text has been checked for correct use of grammar and common technical terms by native English speaking.
Point 2: AOI has been defined in line 91, don’t define it again in line 193. Use its abbreviated form directly.
Answer 2: The revised part marks in blue, as below.
Line 190: tester linked with AOI to detect open, short, and cross short defects on TFT electrode…
Point 3: TFT has been defined in line 29, don’t define it again in lines 42 and 204. Kindly keep harmony with the abbreviation throughout the paper.
Answer 3:
The revision, accordingly. The revised part marks in blue, as below.
Line 42: detecting the in-cell flawed circuit pattern of a TFT, in which a modulator applies…
Line 200: TFT of the 8.5th generation. The dimensions of the glass are 2500 × 2200 mm…
Point 4: The English used in the manuscript is very poor. It is difficult to understand most things. I suggest the authors do an Extensive sentence correction.
Answer 4: The authors express their appreciation for the reviewer's suggestion. The paper has been consulted with support by MDPI publish, which has undergone English language editing. The text has been checked for correct use of grammar and common technical terms by native English speaking.
Point 5: Line 238, it's data line, not dateline. Correct.
Answer 5: The revised part marks in blue, as below.
Line: 234: data line and gate line, the conductivity drives the electronic signal…
Point 6: I am not able to understand why the author uses “which” in the explanations where he/she talks about figure A or B. For instance: “so the output voltage is twice higher in Figure 7, which figure 7 (A)”, “Figure 6, which figure 6 (A) illustrates the array tester”, “n Figure 4, which figure 4 (A) is the foreign material”. It is not correct. Need to perform an English correction.
Answer 6: Thanks for the reviewer's suggestion. The blue indicates the revision, as below.
Line 219: …electro-optical inspector is illustrated in Figure 4; Figure 4A shows foreign
Line 224: …shown in Figure 5A–C; Figure 5A illustrates the topology of optical defect,
Line 241: …output voltage, as shown in Figure 6; Figure 6A illustrates the array tester245: ……
Line 245: …the output voltage is two times higher, as shown in Figure 7; Figure 7A…
Line 249: …therefore the output voltage is more elevated, as shown in Figure 8; Figure 8A…
Line 264: …illustrate the tendency of the circuit characteristics by the array tester, as shown in Figure 9

Round 2
Reviewer 1 Report
I have read the new version. Although the text has been partially improved with respect to the original paper, I feel that some work has still to be done to make it really readable. In the attached file, I have highlighted the most critical points, but more in general I suggest a joint work of the Authors and of the Language Editing Team to improve the form while maintaining the correct meaning of the text.
Some other points are the following ones.
1) Already in the title and then in all the paper, the Authors talk about "Thin-Film Transistor", while they should talk about "Thin-Film Transistor LCD". since a thin-film transistor is only a single switching device.
2) In Eq. (1) the Authors should more clarly write that n is the number of pixels, and in the same way, when the Authors describe Eq. (4), they should talk about "number of pixels" rather than simply "pixel".
3) In all the paper the Authors use the term "task"; the Authors should change this term with a more appropriate one.

Author Response
Dear Editor,
The authors greatly appreciate reviewers' comments and suggestions for our paper. The paper has been revised accordingly. Details of the point-by-point replies are as follows.
Sincerely,
Fu-Ming Tzu
To Reviewer response.
Point 1: In the attached file, I have highlighted the most critical points, but more in general I suggest a joint work of the Authors and of the Language Editing Team to improve the form while maintaining the correct meaning of the text.
Answer 1: The authors express our appreciation for the recommendation by the Reviewer. The paper has been revised according to the review’s comment. The modified sentence marks the blue while maintaining the meaning of the test. Also, the article is checked by the software English Grammarly Premium. The detailed report consists of the essay issues, correctness, clarity, engagement, and delivery through figures 1-5. The result is very positive, and the paper is ready for reference.
- The result of the article is no issue found
- The result of correctness is looking good
- The result of clarity is very clear
- The result of engagement is very engaging
- The result of delivery is just right.
Point 2: Already in the title and then in all the paper, the Authors talk about "Thin-Film Transistor", while they should talk about "Thin-Film Transistor LCD". since a thin-film transistor is only a single switching device.
Answer 2: Revision accordingly. The revised part marks the red.
Point 3: In Eq. (1) the Authors should more clarly write that n is the number of pixels, and in the same way, when the Authors describe Eq. (4), they should talk about "number of pixels" rather than simply "pixel".
Answer 3: Revision accordingly. The revision is as below:
Line 113~136: n represents the pixel quantity in the image. That is a fundamental element to construct the photo. In the experiment, the pixel's dimension depends on the optical resolution and the brightness is concerning the light intensity. The brightness of the pixel decides the grayscale in the image. Line 173~175: The pixel is the smallest unit to constitute a digital image dot matrix; the size depends on the optical resolution: smaller pixel, higher resolution, and vice versa.
Point 4: In all the paper the Authors use the term "task"; the Authors should change this term with a more appropriate one.
Answer 4: Revision accordingly. The authors thank the reviewer's comment. The task was shown in the article has been replaced for either evaluation or experiment. The revised part marks the red.

Round 3
Reviewer 1 Report
In the new version of the paper, some text improvements have been done, even though several sentences remain completely wrong (and, actually, I don't understand how they can pass any logic or grammar test).
Moreover, in several points the substitution of "task" with "experiment", "evaluation" and similar words has not been a great choice (probably it would have been sufficient to use "we").
For example:
1) in the sentence at line 82 "The image was by using a 3 by 3 matrix filter kernel" there is something missing after "was";
2) at line 88 probably you could modify "Therefore, the experiment develops" into "Therefore, we develop"
3) please write better the lines 95-96: "The morphology of the defect shows diversification in TFT-LCD. Notably, the flawed focus on electrical tracking in the case of a functional switch failure."
4) at line 184, replace "presents" with "is presented"
5) at line 186, replace "The evaluation develops" with "We develop"
6) at line 204, replace "the evaluation intends" with "we intend"
7) please improve the sentence at line 231: "The gate line decides the screen resolution that switches the electrical on or off in the alignment layer"
8) at line 276, replace "The evaluation focuses" with "We focus"
9) please correct lines 287-291: "The investigation utilizes the high resolution of the TDI-CCD photosensor to scan the tailor-made defects on a 7-inch quartz mask that verifies the scanning detectability, as shown in Figure 12, in which the scan region in the blue circle is the defects on the peripheral of the photomask to evaluate the optical resolution. As a result, the multiple line scan of the TDI-CCD exhibits the sharpest in the red arrow of the image." They are clearly written in a wrong way.
10) at page 334, replace "the evaluation develops" with "We have developed" or something similar.
I hope the Authors can correct them before publication.
However, at least now the core of the paper is understandable (even though with some difficulties).
Author Response
Point-by-point reply to Reviewers' comments and suggestions
Dear Editor,
The authors greatly appreciate reviewers' comments and suggestions for our paper. The paper has been revised in blue color, accordingly. Details of the point-by-point replies are as follows.
Sincerely,
Fu-Ming Tzu
To Reviewer response.
- in the sentence at line 82 "The image was by using a 3 by 3 matrix filter kernel" there is something missing after "was"
- Thanks for the reviewer. The revision is as below:
The image was constructed by filter kernel using a 33 matrix.
- at line 88 probably you could modify "Therefore, the experiment develops" into "Therefore, we develop"
- Revision done. Therefore, we develop an optoelectronic apparatus with an array tester linked with automatic optical inspection (AOI) to detect the electrical signal on the electrode pixel of the TFT-LCD
- please write better the lines 95-96: "The morphology of the defect shows diversification in TFT-LCD. Notably, the flawed focus on electrical tracking in the case of a functional switch failure."
- Revision done. The morphology of the defects in the TFT-LCD has a diversified type. Notably, the color distortion appears in the display in the case of an abnormal electrical switch.
- at line 184, replace "presents" with "is presented"
- Revision done. The architecture of the optoelectronic apparatus is presented in the next section.
- 5) at line 186, replace "The evaluation develops" with "We develop"
- Revision done. We develop an optoelectronic apparatus, a hybrid array tester linked with AOI
- at line 204, replace "the evaluation intends" with "we intend"
- Revision done. Thus, we intend to develop an alternative solution, a hybrid electrical inspection…
- please improve the sentence at line 231: "The gate line decides the screen resolution that switches the electrical on or off in the alignment layer"
- The gate line controls the electrical current in the alignment layer to decide the screen resolution.
- 8) at line 276, replace "The evaluation focuses" with "We focus"
- Revision accordingly. We focus on the (2) defect area for evaluation
- please correct lines 287-291: "The investigation utilizes the high resolution of the TDI-CCD photosensor to scan the tailor-made defects on a 7-inch quartz mask that verifies the scanning detectability, as shown in Figure 12, in which the scan region in the blue circle is the defects on the peripheral of the photomask to evaluate the optical resolution. As a result, the multiple line scan of the TDI-CCD exhibits the sharpest in the red arrow of the image." They are clearly written in a wrong way.
- Revision accordingly. We utilize the high resolution of the TDI-CCD photosensor to scan the 7-inch quartz mask that verifies the detectability, as shown in Figure 12. The blue circle consists of four defects and is ready for detection: two short defects and two open defects. Consequently, the red arrow illustrates a clear photo.
- at page 334, replace "the evaluation develops" with "We have developed" or something similar.
- Revision accordingly. We have developed a hybrid architecture of the optoelectronic apparatus using an array tester linked with the automatic optical inspection.